# Nanomaterials for Modulating the Aggregation of β-Amyloid Peptides

**DOI:** 10.3390/molecules26144301

**Published:** 2021-07-15

**Authors:** Yaliang Huang, Yong Chang, Lin Liu, Jianxiu Wang

**Affiliations:** 1Hunan Provincial Key Laboratory of Micro & Nano Materials Interface Science, College of Chemistry and Chemical Engineering, Central South University, Changsha 410083, China; 182301010@csu.edu.cn; 2Henan Province of Key Laboratory of New Optoelectronic Functional Materials, College of Chemistry and Chemical Engineering, Anyang Normal University, Anyang 455000, China; 7180610011@stu.jiangnan.edu.cn

**Keywords:** Alzheimer disease’s, amyloid-β, nanomaterials, photothermal therapy, photodynamic therapy

## Abstract

The aberrant aggregation of amyloid-β (Aβ) peptides in the brain has been recognized as the major hallmark of Alzheimer’s disease (AD). Thus, the inhibition and dissociation of Aβ aggregation are believed to be effective therapeutic strategiesforthe prevention and treatment of AD. When integrated with traditional agents and biomolecules, nanomaterials can overcome their intrinsic shortcomings and boost their efficiency via synergistic effects. This article provides an overview of recent efforts to utilize nanomaterials with superior properties to propose effective platforms for AD treatment. The underlying mechanismsthat are involved in modulating Aβ aggregation are discussed. The summary of nanomaterials-based modulation of Aβ aggregation may help researchers to understand the critical roles in therapeutic agents and provide new insight into the exploration of more promising anti-amyloid agents and tactics in AD theranostics.

## 1. Introduction

Abnormal changes in protein spatial structure can lead to the occurrence of protein conformational diseases [1]. For example, the precipitationand aggregation of protein amyloid fibers in neurons or brain parenchyma can induce cytotoxicity and eventually lead to neurodegenerative diseases, such as Alzheimer’s disease (AD), amyotrophic lateral sclerosis (ALS), Huntington’s disease (HD) and Parkinson’s disease (PD) [2,3,4,5]. As the most common form of dementia, AD affects about 40 million people worldwide [6]. Although the pathogenesis of AD is multifactorial [7,8,9,10,11], the abnormal aggregation of β-amyloid (Aβ) peptidesis still considered the most salient feature in AD. Aβ peptides consisting of 39–43 amino acid residues are the proteolytic cleavage products of the β-amyloid precursor protein (APP) [12,13]. Aβ_1–40_ and Aβ_1–42_ are two major abundant types in amyloid plaques [14]. Aβ monomers can assemble into β-sheet-rich oligomers with different sizes and form into long fibrils. Recent studies revealed that extracellular soluble Aβ oligomers and fibrils exhibit strong neurotoxicity, which may be the potential targets for AD treatment [15]. Oxidative stress maybe the potential mechanism that explains the neurotoxicity induced by Aβ aggregates [16]. Thus, the inhibition of Aβ fibrillogenesis and the disassembly of Aβ aggregates are considered to be important treatment strategies for AD.

Prompted by the need to pursue effective treatment of AD, many inhibitors against Aβ aggregation and cytotoxicity were explored, including small molecules, peptides [17,18,19], proteins [20,21,22] and antibodies [23]. For example, we demonstrated that 5,10,15,20-tetrakis(N-methyl-4-pyridyl)-porphyrin (TMPyP), a water-soluble porphyrin, can inhibit Aβ aggregation, disintegrate the preformed Aβ aggregates and alleviate Aβ-induced cytotoxicity [24]. Peptides with a special sequence that is homologous to Aβ can keep it from aggregating through hydrophobic, hydrogen, covalent or electrostatic interactions, which are known as β-sheet breaker peptides, including LPFFD, KLVFFAE, CGGGGGIGLMVG and LVFFARK (LK7). It was suggested that denaturation of native Aβ to oxygenated forms via photooxygenation could slow Aβ aggregation and neurotoxicity [25]. Thus, various photosensitizers were explored for light-induced preclusion of Aβ aggregation, such as methylene blue, porphyrins and riboflavin [26,27,28,29]. However, significant shortcomings limit the further application of these reagents, such as low solubility andpoor stability in physiological conditions. Moreover, the low permeability of the blood-brain barrier (BBB) also renders these inhibitors unsuitable for the treatment of AD [30]. Therefore, the demand is still urgent to develop effective drugs that target Aβ aggregation with great clinical application potential.

With the development of nanotechnology in the past decades, numerous nanomaterials have been designed, synthesized and applied in different fields, such as physics, environmental science and biosensors. Due to the superb biocompatibility, stable physiochemical properties and synthesis and modification flexibility, nanomaterials-based approaches offer enormous potential to overcome the challenges in current therapeutic/diagnostic bio-reagents applications. In this regard, great efforts have been committed to discussing solutions from the nanomaterials perspective for improving AD treatment efficiency. Based on the characteristics of nanomaterials, traditional small molecule inhibitors can get across the BBB by being encapsulated into or modified with nanomaterials, such as mesoporous nano-selenium [31]. Furthermore, nanomaterials can be conjugated with target ligands, such as folate, polysaccharides, cell-penetrating peptidesand antibodies, to improve the bioavailability in brain regions and the efficiency of intracellular particles delivery [32,33]. Although there are a lot of studies that utilized different nanomaterials to inhibit the aggregation of the Aβ peptide, their effects on peptide fibrillation still need to be investigated [34].

In this review, we focused on recent progress in nanomaterials-based methodology for inhibiting Aβ aggregation. Meanwhile, we also paid close attention to novel strategies that use photo-sensitive or enzyme-mimicking properties inAD treatment.

## 2. Nanomaterials

According to the main composition and dimensions, the types of nanomaterialsfor modulating the aggregation of β-amyloid peptidesare various, including gold, carbon, transition oxide, two-dimensional (2D) nanomaterials, metal-organic framework (MOF) and self-assembled nanomaterials (Scheme 1).

### 2.1. Gold-Based Nanomaterials

Gold-based nanomaterials have the strengths ofbeing chemically inert, having tunable local surface plasmon resonance (LSPR) absorption and good conductivity. The LSPR absorption can be tuned by varying the size, shape, surrounding environment and dispersion state [35]. Owing to their unique properties and easeof manipulation, gold-based nanomaterials have broad applications in drug delivery, disease diagnosis and illness treatment, including central nervous system diseases [36]. The main category of gold nanomaterials that are used for treating AD includes bare gold nanoparticles (AuNPs) and gold nanocomposites modified with peptides or other molecules.

AuNPs were reported by many researchers as having functions of penetrating through the BBB, inhibiting Aβ peptide from aggregation [37] and degrading Aβ aggregates based on their size, surface charge, shape, functionality and even concentration [38,39,40,41]. For example, Ma et al. found that the negatively charged citrate-capped AuNPs could induce Aβ peptides to quickly form short protofibrils, subsequently causing them to assemble into short fibril bundles or even bundle conjunctions [42]. Wang and co-workers investigated the effect of AuNPs with different shapes on the aggregation of Aβ_1–40_ peptides (Figure 1A) [43]. Moreover, because of the high degree of surface atomic unsaturation to adsorb Aβ_1–40_ peptides with high affinity, Au nanospheres exhibited a more significant increase of the fibrillation process than Au nanocubes. Liu et al. investigated the size effect of gold nanorods (AuNRs) on modulating the kinetic process of Aβ aggregation (Figure 1B) [44]. They found that the inhibition efficiency of larger AuNRs is better than that of smaller AuNRs and the rate constant was a quadratic function of the diameters or lengths. Liao et al. studied the effect of the surface charge of AuNPs on Aβ aggregation [41]. As shown in Figure 1C, the negatively charged AuNPs could not only inhibit Aβ fibrillization to form fragmented fibrils and spherical oligomers but also remodel preformed fibrils into smaller and ragged Aβ species. Furthermore, by means of enhanced sampling molecular dynamics simulations, the interactions between Aβ peptides and Au nanomaterials with various sizes and morphologies are well characterized, which is helpful for understanding the inhibition mechanism and explore new strategies for AD treatment [45,46].

Small molecules and peptides can be conjugated on the surface of nanomaterials to improve the properties and efficiency of both the modifiers and the nanomaterials [47]. For example, Palmal et al. prepared curcumin-functionalized AuNPs and found that water-soluble AuNPs with multiple curcumin moieties on the surface could inhibit Aβ fibrillation and dissolve Aβ fibrils without using other external agent or force [48]. β-Sheet breaker peptides can also be integrated with Au nanomaterials. Xiong et al. designed a branched dual-inhibitor sequence (VVIACLPFFD) for inhibiting Aβ aggregation and cytotoxicity, whose inhibitory effects were greatly enhanced due to its special surface orientation and conformation (Figure 2A) [49]. However, it was also reported that N-methylated peptides (CGGIGLMVG and CGGGGGIGLMVG) exhibit less effective inhibition of Aβ fibril than free peptides because β-sheet N-methylated peptides, with their more ordered arrangement on the surface, show weak affinity toward Aβ in solution. In contrast, CLPFFD peptides modified on the nanoparticle surface can improve the stability of NPs, reduce the effect on cell viability and increase the delivery efficiency to the brain [50,51]. Moreover, the sequence of the peptide has an influence on the conjugation and stability of AuNPs and the affinity ability for Aβ fibrils [52].

Aβ aggregation can be disintegrated via local heat generation. Kogan et al. found that AuNPs modified with Cys-PEP peptides and CLPFFD can selectively attach to the Aβ fibrils and prefibrillar intermediate amyloidogenic aggregates (PIAA) (Figure 2B) [53,54]. Then, microwave irradiation generated local heat using nanoparticles to dissolve Aβ aggregates. In photothermal therapy (PTT), light can be converted into heat with the aid of molecules or nanomaterialsthat display photothermal conversion ability. PTT was shown to be a promising strategy for the treatment of various diseases due to its remarkable advantages of site- and time-specificity, non-invasiveness and targeting selectivity. Owing to the extinction coefficient of theLSPR, Au nanomaterials can cause the photothermal ablation of amyloid peptide aggregates using laser irradiation [50,55,56]. As shown in Figure 2C, using penetratin peptide (Pen)-modified Au nanostars as part of an NIR photothermal method could be activated using ultralow irradiation to treat AD [57]. Due to the irregular morphology, Au nanostars possessed a high NIR absorption-scattering ratio and large specific surface area. Pen peptides can enhance the travel across the BBB and the cellular internalization of nanomaterials. The Pen-Au nanostars not only inhibited the formation of Aβ fibrils but also disassembled the preformed Aβ fibrils under the NIR irradiation. Moreover, a fluorescent ruthenium complex was loaded on the nanostars for tracking the drug delivery. To shorten the laser irradiation time, Lin et al. applied an NIR femtosecond laser for the destruction of the preformed AuNR-modified Aβ fibrils (Aβ fibrils@AuNRs) (Figure 2D) [58]. The results showed that in the presence of AuNRs, the femtosecondlaser irradiation could efficiently dissociate the Aβ fibrils into small fragments with a non-β-sheet structure in 5 min at a safe energy level and the morphology of AuNRs was transformed into amorphous shapes. However, there was no obvious destruction effect on the Aβ fibrils or Aβ fibrils@Au nanospheres under the laser irradiation.

Metal ions, such as Zn^2+^ and Cu^2+^, were shown to participate in the pathology of AD [59,60]. A metal chelator can capture metal ions, hamper ROS formation and inhibit metal ion-induced Aβ aggregation [61]. However, they have some disadvantages, such as poor permeability of the BBB and limited ability to distinguish toxic metal ions related to Aβ aggregates from those associated with normal biological homeostasis [62,63]. Inspired by stimuli-responsive controlled-release drug delivery systems, the controllable release of chelators from “containers” may avoid this problem. Shi et al. reported a dual-responsive “caged metal chelator” release system based on NIR-absorbing Au nanocages [64]. As illustrated in Figure 3A, the chelator of clioquinol (CQ) was encapsulated in Au nanocages and the pore was blocked by human IgG via the redox- and thermal-sensitive arylboronic ester bond. Over-produced H_2_O_2_that was induced by deviant Aβ-metal ions aggregates would initiate the degradation of arylboronic ester and the subsequent release of CQ. Moreover, Au nanocages with NIR light could generate local heat to break the bond, thus enhancingthe CQ release and dissolving Aβ deposits via noninvasive remote control.

Unlike AuNPs, gold nanoclusters (AuNCs) consist of several to dozens of atoms that have excellent optical properties of intense fluorescence and high photostability. As shown in Figure 3B, Hao et al. found that AuNCs modified with CLVFFA via Au-S bonds showed improved inhibitory ability [65]. The results exhibited that AuNCs-CLVFFA could block the fibrillogenesis of Aβ_1–40_ and the prolongation of fibrils and disaggregate the mature fibrils into oligomers. Zhang et al. found that Cys-Arg (CR) dipeptide-caped Au_23_(CR)_14_ NCs could completely dissolve exogenous mature Aβ fibrils and endogenous Aβ plaques and restore the natural unfolded state of Aβ peptide from a β-sheet structure [66].

### 2.2. Carbon

Carbon nanomaterials, including carbon nanotubes, graphene, fullerene, carbon nanospheres and carbon dots, have received great attention in the biological field due to their unique physical and chemical features. Carbon nanomaterials with hydrophobic surfaces can interact with various biomolecules, such as DNA, proteins and amyloid nanostructures. Moreover, when entering into a living organism, they may disturb the self-assembly processes of peptides or proteins.

#### 2.2.1. Graphene-Based Materials

Owing to the large available surface and hydropathy, graphene oxide (GO) was applied to modulate the aggregation of Aβ via the adsorption of amyloid monomers and decreasing the kinetic reaction [67,68,69]. Yang et al. found that pristine graphene and GO could inhibit Aβ peptide fibrillation and clear mature amyloid fibrils through experimental and computational investigation [70]. Mahmoudi et al. found that the formation of a protective protein corona on GO sheets could further enhance the inhibition effect [71]. The size effect of GO on modulating Aβ aggregation was also investigated by Wang and co-workers [72]. Surface chirality was also shown to play an important role in protein adsorption dynamics and cell behaviors. As shown in Figure 4A, Qing et al. studied the chiral effect on amyloid formation by using cysteine-enantiomer-modified GO as a platform [73]. The result showed that *R*-cysteine-modified GO suppressed the absorption, nucleation and fiber elongation processes of Aβ_1–40_, thus leading to a remarkable inhibition rate of amyloid fibril formation. However, *s*-cysteine-modified GO accelerated these processes. The stereoselective interaction between chiral moieties and Aβ peptides caused the enrichment of oligomers on the GO surface, but the distance between them should be short enough (1–2 nm). This work provided novel insights into understanding the key roles of biological membranes on protein amyloidosis.

Based on its high optical absorption in the NIR region, graphene has been widely explored in biomedical applications. In 2012, Li et al. first reported the photothermal treatment of AD by using thioflavin-S (ThS)-modified GO (Figure 4B) [74]. The ThS-modified GO can produce local heat to dissociate Aβ fibrils under low-power NIR laser irradiation with improved selectivity because of the specific targeting of ThS toward amyloid. Moreover, the Aβ morphology change during the photothermal treatment can be monitored in real time by recording the increased fluorescence of ThS in the complex of ThS and Aβ fibrils. Taking advantage of the permeation and disruption of the cellular membrane by Aβ oligomers, Xu et al. developed an oligomer-self-triggered and NIR-enhanced system based on the lipid-bilayer-coated graphene (GMS-Lip) [75]. Dyes and drugs were co-loaded within GMS-Lip, which could be released by the amyloid oligomers. In addition, thermal-sensitive Lip could be further destroyed by the heat produced by graphene under an NIR laser to ensure the release efficiency. Moreover, the NIR-assisted release could be initiated locally by tracking the fluorescence.

GO can hybridize with other nanomaterials to achieve improved inhibition performance based on the synergistic effect. Moreover, the decoration of nanomaterials on GO can avoid the easy agglomeration and restacking of GO. For instance, GO/AuNPs nanocompositeswere prepared using pulsed laser ablationin water to reduce Aβ aggregation, which produced an enhanced effect compared with using GO or AuNPsalone [76]. Due to the hydrophobic interactions between the nanocomposites with hydrophobic amino acids of Aβ_1–42_, the nucleation process was significantly disturbed and the fibrillation was subsequently slowed. The result showed that the GO/AuNPs nanocomposites not only reduced Aβ_1–42_ aggregation and cytotoxicity but also led to the deploymerization of amyloid fibrils and inhibition of their cellular cytotoxicity. In addition, Ahmad et al. reported that a GO–Fe_4_O_4_ nanocomposite showed the enhanced inhibitory effect of Aβ_1–42_ peptides and depolymerized Aβ_1–42_ fibrils (Figure 4C) [77]. 

#### 2.2.2. Carbon Nanotubes (CNTs)

CNTs are tubular structures of rolled-up sheets of graphene, including single-wall (SWCNTs) and multi-wall (MWCNTs) species. CNTs can be exploited for biosensing, tissue engineering and drug delivery due to their chemical stability, good porosity and high surface area. Moreover, SWCNTs were shown to shuttle drugs into a wide range of cell types [79,80]. Luo et al. found that SWNTs can induce Aβ peptides to form β-sheet-rich yet non-amyloid fibrils, and Aβ peptidescan reduce the toxicity of SWNTs [80]. However, the poor dispersibility of pristine SWNTs in solution greatly decreased the inhibition efficiency. Functionalization of SWNTs with hydrophilic groups may increase their dispersibility. Liu et al. investigated the inhibitory effect of SWNT-OH on Aβ_1–42_ fibrillogenesis [78]. As shown in Figure 4D, the percentage of hydroxyl groups in SWNT-OH was crucial for their inhibition capacity against Aβ_1–42_ aggregation. Moreover, SWNT-OH could transform the mature fibrils into smaller granular aggregates but not oligomers. Xie et al. revealed that the electrostatic, hydrophobic and aromatic stacking interactions between hydroxylated SWCNT and Aβ_16–22_ not only inhibit the Aβ_16–22_ fibrillization but also shifted the conformations of oligomers from the orderedβ-sheet-rich structures into the disordered coil aggregates [81].

#### 2.2.3. Carbon Nanospheres

Phototherapy, including photothermal therapy (PTT) and photodynamic therapy (PDT), is generally activated by visible or first NIR-I light. However, it has limited tissue penetration through dense skull and scalp and may cause damage to nearby normal tissues. Thus, utilizing excitation light at second near-infrared light (NIR-II, 1000–1700 nm) is a more attractive option for deeper tissue penetration and a lower signal-to-noise ratio. Ma et al. designed Aβ targeting, N-doped three-dimensional mesoporous carbon nanospheres (KD8@N-MCNs) for NIR-II PTT of AD (Figure 5A) [82]. KLVFFAED (KD8) was used as the target of Aβ and receptor of the advanced glycation end-products (AGEs). N-MCN was selected as the NIR-II photothermal agent due to its excellent photothermal effect and superoxide dismutase (SOD) and catalase (CAT)-like activities. Combining the above advantages, KD8@N-MCNs can dissolve Aβ_1–42_ aggregates under NIR-II illumination, scavenge intracellular ROS and alleviate neuroinflammation in vivo. Moreover, KD8@N-MCNs can efficiently cross the BBB due tothe modification of KD8 on the nanosphere’s surface.

#### 2.2.4. Fullerene

Fullerene possesses the advantages of strong hydrophobicity and high electrophilicity, which endow it with the potential ability to inhibit the aggregation of Aβ peptides [83]. For example, Xie et al. investigated the molecular mechanism of fullerene-based inhibition, which could be attributed to the strong hydrophobic and aromatic-stacking interactions between the fullerene hexagonal rings and the Phe rings of Aβ_16–22_ peptide [84]. Moreover, Kim et al. reported that a C_60_ derivate (1,2-dimethoxymethanofullerene) could bind to the central motif, namely, Aβ_16–20_ (KLVFF), based on the strong hydrophobic interaction and inhibit the aggregation of Aβ peptides at the early stage [85]. After that, other C_60_ derivates were also synthesized and applied for inhibiting the aggregation of Aβ peptides, including hydroxylated fullerene and sodium fullerenolate Na_4_[C_60_(OH)_~30_] [86,87,88,89,90]. 

C_60_ could produce reactive oxygen species (ROS) viathe photo-excitation of C_60_ and O_2_ to stimulate DNA and protein photocleavage [91,92]. Ishida et al. employed a C_60_–sugar hybrid to inhibit Aβ aggregation and degrade Aβoligomers under long-wavelength UV radiation and neutral conditions [93]. Besides the hydrophobic interaction between Aβ and C_60_, the hydrophilic sugar can interact with the hydrophilic N-terminal of Aβ_1–42_ through the formation of hydrogen bonds. Aiming to improve the solubility, C_60_ was functionalized with a sulfo or amino group [94]. These two hydrophilic groups could further strongly interact with the termini of Aβ_1–42_through the ionic interactions and/or hydrogen bonds.

In addition to generating ROS under photo-excitation, C_60_ can be used as a ROS scavenger in the dark. Du et al. designed a NIR-switchable nanoplatform for synergy therapy of AD based on the ROS-generating and -quenching properties of C_60_ [95]. As shown in Figure 5B, UCNPs were conjugated with C_60_ and the Aβ-target peptide KLVFF. Under NIR irradiation, C_60_ was photo-sensitized using UCNPs through FRET to produce ROS, leading to the oxygenation of Aβ and inhibition of its aggregation. In the dark, C_60_ could eliminate the overproduced ROS to protect the cell from oxidative stress.

#### 2.2.5. Carbon Dots

Since being accidentally discovered as byproducts during the purification of SWCNTs, carbon dots (CDs) have received growing interest in biomedical and biosensing fields [96]. As a potential alternative to semiconductor QDs, CDs have prominent characteristics, such as their low cost, ease of synthesis, good biocompatibility and intrinsic fluorescence. Moreover, the absorption and emission spectra can be tuned by adjusting the degree of carbonization and the percentage of surface moieties.

CDs generally have abundant functional groups on their conjugated aromatic core, such as hydroxyl, amino and carboxyl groups. They can interact with Aβ peptides and aggregates through electrostatic, hydrogen bonding, π-π stacking and hydrophobic interactions because of their abundant surface chemical properties and small size (generally less than 10 nm). Liu et al. demonstrated that graphene quantum dots (GQDs) can inhibit Aβ_1–42_ peptide aggregation, mainly via hydrophobic interactions, and rescue Aβ_1–42_ oligomer-induced cytotoxicity [97]. The surface chirality of nanoparticles also has an influence on the inhibition efficiency against Aβ aggregation [98]. L-Lys-CDs were reported to exhibit a higher affinity toward Aβ_1–42_ peptides than D-Lys-CDs and could remodel the Aβ_1–42_ secondary structure and fibril morphologies [99]. CDs can cross the BBB because of their small size, which allows them to be utilized as nano-carriers to transport functional molecules to the brain. Thus, GQDs conjugated with an endogenous neuroprotective glycine-proline-glutamate peptide showed an enhanced neuroprotective effect and improved learning and memory capability of APP/PS1 mice [100]. CDs can also be used as fluorescent nanoprobes for biological imaging. Among them, CDs with red fluorescence possess several advantages of low biological fluorescence background signal and high tissue penetration ability. Recently, Gao et al. discovered new functions of nitrogen-doped carbonized polymer dots (CPDs) to target Aβ aggregation [101]. The non-covalent interactions between Aβ aggregates and CPDs limited the molecular vibration and rotation of CPDs, resulting in the red emission. As shown in Figure 6A, after interacting with Aβ fibrils, CPDs emitted an increased red fluorescence signal, indicating that CPDs can be utilized as a multifunctional therapeutic agent for disintegrating and monitoring Aβ fibrils.

Photoexcited CDs can generate reactive oxygen species (ROS) through energy-transfer and electron-transfer pathways, which endows them with the ability to denature toxic biomolecules. For example, branched polyethylenimine-passivated CDs (bPEI@CDs) were reported as photosensitizers that inhibit the self-assembly of Aβ peptide and disassemble preformed Aβ aggregates upon light irradiation (Figure 6B) [102]. Photoactivated bPEI@CDs produced ROS to oxygenate and break β-sheet-rich Aβ aggregates into smaller fragments. Meanwhile, it was more efficient for reducing Aβ-mediated toxicity compared with bPEI@CDs under dark conditions. However, the UV-to-Vis dependent photoexcitation is a potential obstacle to the use of bPEI@CDs. To improve the effectiveness of photoexcitation of CDs and grant the ability to target Aβ, red-light-responsive DNA aptamer-functionalized CDs (Apta@CDs) were designed for the Aβ-targeting spatiotemporal suppression of Aβ aggregation [103]. As shown in Figure 6C, Apta@CDs could specifically bind to Aβ aggregates in the 5xFAD (five familial Azheimer’s) mouse brain. Photoexcited Apta@CDs under red LED (617 nm) light can generate ^1^O_2_, denature the Aβ peptide, slow the formation of β-sheet-rich aggregates and alleviate the Aβ-associated cytotoxicity.

### 2.3. Metal-Oxide Nanomaterials

#### 2.3.1. Magnetic Nanoparticle (MNPs)

MNPs have the advantages of good biocompatibility and unique magnetic properties. Theywere deemed as therapeutics and imaging agents inthe treatment of brain diseases. Mahmoudi et al. investigated the physicochemical effects (size, charge and surface treatment) of coated superparamagnetic iron oxide nanoparticles (SPIONs) on Aβ aggregation (Figure 7A) [104]. They found that the size and surface area have significant effects on Aβ fibrillation. Lower concentrations of SPIONs inhibited the fibrillation and higher concentrations promoted the rate, reversely. Furthermore, the positively charged SPIONs promoted fibrillation at significantly lower particle concentrations. In addition, peptide- and antibody-modified MNPs also significantly inhibited Aβ fibrillation [105,106].

MNP-based targeted tissue drug delivery was shown to be a promising therapeutic approach because MNPs can be directed toward a specific site in disease tissue locations by a magnetic force. Mesoporous silica-coated MNPs (MMSNPs) were utilized as smart vehicles to encapsulate quercetin (QC) with anti-amyloid and antioxidant properties, overcoming the limitations of poor solubility and bioavailability (Figure 7B) [107]. They found that biophenols QC can bind with Aβ monomers and oligomers to block the fibril formation [108]. Moreover, the released QCs could decrease the Aβ-related cytotoxicity and minimize the Aβ-induced ROS. Li et al. designed light-responsive magnetic nanoparticle prochelator conjugates for inhibiting metal-induced Aβ aggregation [109]. In this complex, Fe_3_O_4_ NPs were utilized to cage CQ through a photoactive o-nitrobenzyl bromide (ONB) linkage and the conjugates did not interact with Cu^2+^ in the prochelator form. After the photolytic cleavage under UV radiation at 365 nm, the caged CQ was liberated from the NPs, efficiently preventing metal-induced Aβ aggregation, decreasing cellular reactive oxygen species (ROS) and protecting cells against Aβ-related toxicity.

Previous studies indicated that MNPs can effectively improve the local temperature in the area extremely close to the MNPs (about 5 nm) through alternating the magnetic field, which reduces the penetration depth limit and damage to the neighboring tissues [111,112]. Therefore, it is a non-invasive strategy for clearing Aβ aggregates by using MNPs/AMF hyperthermia [113]. For instance, Loynachan et al. used LPFFD-functionalized PEG-coated MNPs forthe remote magnetothermal disruption of Aβ aggregates under a high-frequency alternating magnetic field (AMF) (Figure 7C) [110]. They found that the local heat dissipated by targeted MNPs could dramatically decrease the size of Aβ aggregates from microns to tens of nanometers upon exposure to a physiologically safe AMF, which isattributed to the dissociation of Aβ deposits. The reduced Aβ cytotoxicity due to the magnetothermal disruption was confirmed in primary hippocampal neuronal cultures. Moreover, for simultaneous diagnosis and treatment, an Aβ oligomer-sensitive naphthalimide-based fluorescent probe (NFP) was loaded on the KLVFF-conjugated MNPs [114].

#### 2.3.2. Polyoxometalates (POMs)

Polyoxometalates (POMs) are early-transition-metal-oxygen-anion clusters that usually include the d^0^ species V(V), Nb(V), Ta(V), Mo(VI) and W(VI) with versatile structures. They have been widely explored in recent years for biomedical applications [115] and showed remarkable effects against acquired immune deficiency syndrome (AIDS) [116]. Because of the similarity to water-solubilized fullerene derivatives, POMs can be utilized as Aβ aggregation inhibitors. For example, Li et al. identified four POMs with the size-dependent ability of inhibiting Aβ peptide aggregation via a high-throughput screening method, in which K_8_[P_2_CoW_17_O_61_] with a Wells–Dawson structure exhibits the highest inhibition (Figure 8A) [117]. They also found that POMs electrostatically bind to the cationic His_13_–Lys_16_ cluster (HHQK) of Aβ peptides. Zhou et al. demonstrated that two POMs with the wheel-shaped Preyssler structure and the Keggin-type structure could interact with Aβ_1–40_ and inhibit its fibrillization [118]. To suppress the peroxidase-like activity of Aβ-hemin, POM-Dawson was further functionalized with transition metal ions of various histidine-chelating metals. These transition metalPOMs not only specifically targeted the HHQK in Aβ peptides but also showed a stronger inhibition effect on Aβ-hemin formation [119]. After that, organoplatinum-substituted POMs were reported to inhibit Aβ aggregation [120]. Moreover, POMs can also significantly inhibit metal-ion-induced Aβ_1–40_ aggregation because POMs with high negative charges show strong interactions with Zn^2+^ and Cu^2+^ [121].

POMs have been broadly used for homogeneous photocatalysis in water to produce ROS and oxidize various substrates, including pyrimidine bases [122,123]. Inspired by this fact, Li et al. demonstrated that POMs can photodegrade Aβ monomers and even oligomers under photoirradiation conditions [124]. At a low concentration, POMs still showed ahigh inhibition efficiency under UV irradiation. To disaggregate the neurotoxic Aβ fibrils, Li et al. further developed a redox-activated NIR-responsive reduced POMs (rPOM)-based agent for the photothermal treatment of AD (Figure 8B) [125]. In this work, mesoporous silica nanoparticles (MSNs) were used to load rPOM and thermally responsive copolymer poly(N-isopropylacrylamide-co-acrylamide) was employed to cap the pores of MSNs to prevent rPOMs leakage and ensure they remained intact. When being irradiated by an 808 nm laser, rPOMs with strong NIR absorption produced local hyperthermia to melt the shell away from the channels, which would inhibit Aβ aggregation and disaggregate Aβ fibrils by local heat. Moreover, rPOM could act as antioxidants to clear ROS and the product of POMs can inhibit Aβ aggregation.

POMs can be employed as inorganic nanobuilding blocks to fabricate organic–inorganic assembly nanoarchitectures with the aid of peptide or protein building blocks. POM–peptide (POM@P) hybrid particles were synthesized as two-in-one bifunctional particles through the self-assembly of these dual inhibitors (Figure 8C) [126]. Aβ_15–20_ peptides (Ac–QKLVFF–NH_2_) with a high local density can not only bind the homologous sequence in Aβ peptides and disrupt their aggregation but also enhance the targeting inhibition efficiency. Moreover, congo red (CR), which is a clinically used Aβ fibril-specific staining dye, was loaded into the nanospheres for real-time monitoring of the inhibition process of POM@P. Todevelop novel drugs with multiple functions against AD, Gao et al. synthesized AuNPs@POM–peptide as a novel multifunctional Aβ inhibitor (Figure 8D) [127]. In this hybrid, AuNPs were utilized to carry two inhibitors (POMs and LPFFD peptides) to efficiently cross the BBB. In addition to the inhibition of Aβ aggregation and the dissociation of Aβ fibrils, AuNPs@POMD–peptides also decreased Aβ–heme peroxidase activity and Aβ-induced cytotoxicity via synergistic effects. Based on a similar principle, AuNRs with a high NIR absorption property were used instead of AuNPs for the hyperthermia-induced disassembly of Aβ fibrils [128]. Moreover, due to the shape and size-dependent optical properties, AuNRs could also be used to sensitively detect the Aβ aggregates.

#### 2.3.3. Cerium Oxide Nanoparticles (CNPs)

Owing to their rapid transformation between Ce^4+^ and Ce^3+^ at physiological pH, CeO_2_ NPs can act as free radical scavengers and protect cells from oxidative stress. CeO_2_ NPs possess powerful multienzyme activity, including superoxide oxidase, catalase and oxidase [129]. Although CeO_2_ NPs exhibit no obvious inhibition effect on Aβ aggregation, it is usually utilized to eliminate over-expressed ROS produced by Aβ-Cu^2+^ and protect against neurodegeneration [130,131]. To realize the targeted delivery of AD therapeutic agents, Li et al. designed a double delivery platform for AD treatment based onthe H_2_O_2_-responsive release of antioxidant CeO_2_ NPs and the metal chelator clioquinol (CQ) [132]. In this study, CQ was entrapped into the phenylboronic-acid-modified MSN (MSN-BA) and glucose-coated CeO_2_wasused as the gatekeeper via the formation of cyclic boronate moieties. H_2_O_2_ could induce the breakage of arylboronic esters, thus resulting in the release of CeO_2_ NPs and CQ. Finally, the two-in-one bifunctional nanoparticles effectively inhibited Aβ aggregation, reduced intracellular ROS and rescued cells from Aβ-related toxicity. To treat AD in multiple pathways, Guan et al. prepared CeO_2_ NPs with a functional MnMoS_4_ shell (CeNPs@MnMoS_4_-n; *n* = 1–5), as displayed in Figure 9A [133]. MnMoS_4_ could eliminate intracellular toxic Cu^2+^ through ion exchange. In turn, the release of Mn^2+^ promoted neurite outgrowth. Moreover, CeNPs@MnMoS_4_-3 with the SOD activity decreased the oxidative stress.

According to previous reports, POMs as a class of artificial proteases have the ability to hydrolyze peptides [134]. Cerium dioxide/POMs (CeONP@POMs) mixed nanoparticles were used to mimic metallopeptidase for the treatment of the neurotoxicity caused by Aβ (Figure 9B) [135]. The mixed NPs with peptide hydrolysis and superoxide dismutase activity efficiently degraded Aβ aggregates and decreased cellular ROS. Moreover, in addition to promoting PC12 cell proliferation and crossingthe BBB, CeONP@POMs inhibited Aβ-induced BV2 microglial cell activation. Kim et al. designed an extracorporeal strategy for cleansing blood Aβ by using core/shell structured multifunctional magnetite/ceria nanoparticle assemblies (MCNAs) [136]. As shown in Figure 9C, the nano-assemblies were further conjugated with antifouling polyethylene glycol (PEG) and Aβ antibodies for the specific capture of Aβ peptides. The MNPs enabled the magnetic separation of the captured Aβ peptides by applying an external magnetic field. The ceria NPs alleviated oxidative stress by scavenging the ROS generated by the immune response during the process. The blood-cleansing treatment of 5xFAD transgenic mice demonstrated that the levels of Aβin the blood and brain were effectively reduced and the spatial working memory deficit was rescued.

### 2.4. 2D Nanosheets

#### 2.4.1. Black Phosphorus

As a new member of 2D layered semiconductor nanomaterials, black phosphorus (BP) has attracted broad attention because of its perfect optical and thermal properties. Moreover, BP can degrade into nontoxic phosphate and phosphite anions under physiological conditions. In 2019, Lim et al. synthesized titanium sulfonate ligand (TiL_4_)-modified BP nanosheets and BP quantum dots and reported for the first time that they could inhibit Aβ_1–40_ to form fibrils by adsorbing the monomers [137]. To further improve the stability of BP, Yang et al. constructed an inhibitor (LK7)-coupled and polyethylene glycol (PEG)-stabilized BP-based nano-system (PEG-LK7@BP) (Figure 10A) [138]. Besides the enhanced electrostatic and hydrophobic interactions, LK7 with a high local concentration on the BP surface could enhance the affinity between the Aβ species and the BP. Based on the synergistic effect, PEG-LK7@BP prevented the conformational shift of Aβ_1–42_ to a β-sheet structure, suppressed Aβ_1–42_ aggregation and attenuated the toxicity of Aβ_1–42_.

BP with a thickness-dependent energy bandgap from 0.3 eV (bulk) to 2.0 eV (monolayer) has broad absorption across the ultraviolet and entire visible light regions. BP, including the bulk material and ultrathin nanosheets, was found to be an effective photosensitizer for the generation of ^1^O_2_ under the entire visible light region and was applied in the fields of catalysis and PDT [139]. Interestingly, BP QDs, BP NPs and BP NSs have been reported to possess NIR photothermal properties for PTT [140,141,142]. Li et al. employed BP nanosheets as the NIR-activated photosensitizer to generate ^1^O_2_ and oxidize Aβ peptide in vitro and in vivo (Figure 10B) [143]. BP nanosheets were modified with 4-(6-methyl-1,3-benzothiazol-2-yl) phenylamine (BTA) to increase the stability and endowed BP with Aβ binding selectivity. Moreover, the BP@BTA could reduce the Aβ-induced cytotoxicity and show neuroprotection to the transgenic strain Caenorhabditis elegans CL2006.

#### 2.4.2. Transition Metal Dichalcogenides

Transition metal dichalcogenides (TMDs) have attracted worldwide attention in the areas of nanoelectronics, optoelectronics and electrocatalysis. They are always used for drug delivery and tissue ablation. The basal plane of TMD NSs can adsorb or conjugate various aromatics (e.g., pyridine and purine) and other compounds. Mudedla et al. investigated the interaction between Aβ fibrils and MoS_2_-based materials and found that MoS_2_ nanotubes could inhibit the aggregation of smaller protofibrils to matured fibrils and bust the preformed fibrils [144]. Wang et al. confirmed the inhibition effect of monolayer MoS_2_ on the Aβ_33–42_aggregation [145]. Liu et al. reported the concentration-dependent contradictory effect of AuNP-decorated MoS_2_ nanocomposites on Aβ_1–40_ aggregation [146]. As displayed in Figure 11A, a low concentration of AuNP-MoS_2_ nanocomposite could act as the nuclei to accelerate the nucleus formation and fibrillation of Aβ_1–40_, but a high concentration of nanocomposites could limit the structural flexibility of Aβ_1–40_, leading to the inhibition of nucleus formation and aggregation.

The 2D TMDs analogous to graphene exhibit excellent properties, such as high colloidal stability in aqueous media and a high mass extinction coefficient at 800 nm [147]. Li et al. first reported that WS_2_ could adsorb Aβ_1–40_ monomers on the surface through van der Waals and electrostatic interactions, effectively inhibiting Aβ aggregation and dissociating the preformed Aβ fibrils via photothermal ablation upon NIR irradiation [148]. To further enhance the inhibition ability, Wang et al. prepared multifunctional MoS_2_/AuNR nanocomposites with high stability and good biocompatibility through electrostatic self-assembly [149]. This nanocomposite with high NIR absorption can modulate the aggregation of Aβ peptides, disrupt mature fibrils under low laser power NIR irradiation and alleviate Aβ-induced ROS against neurotoxicity. 

Artificial Aβ-degrading enzymes were designed for the efficient cleavage of Aβ [150,151]. However, the specific hydrolysis sites are always embedded inside the β-sheet structure, hindering the access and hydrolysis efficiency. Therefore, Ma et al. developed a NIR (near-IR) controllable artificial metalloprotease (MoS_2_–Co), combining MoS_2_ and a cobalt complex of 1,4,7,10-tetraazacyclododecane-1,4,7,10-tetraacetic acid (Codota) (Figure 11B) [152]. MoS_2_–Co can inhibit the formation of a β-sheet structure and shorten the distance between Aβ peptides and MoS_2_–Co. Moreover, under NIR irradiation, MoS_2_–Co can produce local heat to disintegrate Aβ aggregates and facilitate the hydrolysis activity of Codota toward Aβ peptides.

MoS_2_ NPs are also drawing more and more interest as self-lubricating coatings and in biochemical applications. Han et al. prepared spherical polyvinylpyrrolidone-functionalized MoS_2_ NPs with an average diameter of 100 nm using a pulsed laser ablation method and found that they show multifunctional effects on Aβ_1–42_ aggregation (Figure 11C) [153]. MoS_2_ NPs could adsorb Aβ_1–42_ monomers or oligomers on the surface based on the interaction between MoS_2_ NPs and the hydrophobic region of Aβ_1–42_. This delays the nucleation process, inhibits Aβ_1–42_aggregation and destabilizes the preformed fibrils. As a result, the calcium channel induced by the incorporation of Aβ_1–42_ oligomers into neuronal cell membranes was blocked to ensure calcium homeostasis and protect neuronal cells. Moreover, MoS_2_ NPs could reduce the intracellular ROS (·OH) level induced by Aβ_1–42_.

#### 2.4.3. Graphitic Carbon Nitride

Unlike bulk g-C_3_N_4_, the 2D ultrathin g-C_3_N_4_nanosheet is the most stable allotrope of carbon nitride under ambient conditions. It has excellent properties of good biocompatibility, high surface-area-to-volume ratio and nontoxicity and was employed in bioimaging, drug delivery and cancer diagnosis.

A g-C_3_N_4_nanosheet, with a narrow band gap of 2.7 eV, can act as a stable photocatalyst for water splitting and the degradation of organic pollutants. Chung et al. applied photoactive g-C_3_N_4_ for the light-induced suppression of Aβ aggregation and toxicity [154]. As shown in Figure 12A, the photosensitized g-C_3_N_4_ generated oxidative ROS through photoinduced electron transfer under visible-light illumination, which further oxidized Aβ peptides, preventing the aggregation of Aβ monomers. However, g-C_3_N_4_ had no obvious effect on Aβ aggregation under dark conditions. Moreover, doping transition metal ions could promote ROS generation and enhance their inhibition efficiency. To enhance the photodegradation efficiency, Wang et al. used GO/g-C_3_N_4_ as the photocatalyst for irreversible disassembly of Aβ_33–42_ aggregate into nontoxic monomers under UV (Figure 12B) [155]. In this nanocomposite, GO acts as an Aβ collector due to its high surface area and abundant functional groups. g-C_3_N_4_ was also decorated with AuNPs to separate photoexcited electron-hole pairs [156]. 

Based on its strong adsorption capacity for metal ions, Li et al. reported that a g-C_3_N_4_nanosheet could act as the chelator to block Cu^2+^-induced Aβ aggregation, disaggregate the preformed Aβ-Cu^2+^ aggregates, reduce the ROS level induced by Aβ-Cu^2+^ and block Aβ-mediated toxicity [157]. At the same time, they also found that platinum(II)-coordinated g-C_3_N_4_ (g-C_3_N_4_@Pt) can covalently bind to Aβ monomers and oxygenate Aβ monomers and oligomers upon visible light irradiation, thus inhibiting the aggregation and toxicity of Aβ [158]. Furthermore, they found that the accumulation of oxygenated Aβ can inhibit the aggregation of native Aβ peptides.

For targeted therapy, Gong et al. developed an intelligent Aβ nanocaptor by anchoring C_3_N_4_ nanodots to Fe_3_O_4_@MSNs and modifying them with benzothiazole aniline (BTA) (B-FeCN), as shown in Figure 12C [159]. In this nanocomposite, C_3_N_4_ nanodots could capture Cu^2+^, subsequently blocking the formation of the Aβ-Cu^2+^ complex and diminishing Aβ aggregation. Fe_3_O_4_ could cause local low-temperature hyperthermia to enhance the BBB permeability and dissolute the Aβ plaques. In addition, BTA endowed the nanocaptor with aspecific targeting ability and fluorescent imaging property for monitoring Aβ aggregates.

### 2.5. Metal-Organic Frameworks

As increasingly popular crystalline porous materials, metal-organic frameworks (MOFs) are built from metal nodes and organic linkers. Benefitting from the control of chemical functionality, pore size and crystal morphology, they have been used in many fields, including catalysis, gas storage, drug delivery and biosensing. Owing to the exposed metal sites and porphyrin linkers with aromatic rings, porphyrinic MOFs are particularly attractive for biomedical research. Porphyrinic MOF PCN-224 was prepared for the NIR-induced suppression of Aβ peptide aggregation (Figure 13A) [160]. Besides good biocompatibility and excellent stability, PCN-224 showed singlet oxygen generation capability in the NIR window because of the high density of the photosensitizer molecule TCPP in the framework and the easy diffusion of O_2_ through the porous structure. The results showed that the photoactivated PCN-224 could effectively inhibit the aggregation of Aβ_1–42_ into a high-order β-sheet-rich structure and rescue the cytotoxicity of Aβ_1–42_. According to the previous report, the nitrogen atoms in porphyrin from porphyrinic MOF possess a high binding affinity to Cu^2+^ ions [161]. Inspired by these findings, Yu et al. utilized porphyrinic MOF as a Cu^2+^-chelator and photooxidation agent for inhibiting Aβ peptide aggregation (Figure 13B) [162]. To further enhance the selectivity and photooxidation efficiency, MOFs were modified with the Aβ-targeting peptide LPFFD. As one sub-class of the MOF family, Prussian blue (PB) has numerous applications, including electrocatalysis, bioimaging and biosensing. According to the previous reports, it can act as a nanozyme to scavenge ROS and trap metal ions in its lattice cavities. Recently, Kowalczyk et al. studied the dual effects of PB on inhibiting Aβ_1–40_ aggregation and chelating Cu^2+^ [163]. They found that PBcould accelerate the nucleation of Aβ_1–40_ and facilitate the formation of Aβ_1–40_ amorphous aggregates instead of β-sheet fibrils. 

### 2.6. Semiconductor Quantum Dots

Unlike traditional fluorescent organic dyes, semiconductor quantum dots (QDs) have shown excellent optical properties, including size-dependent emission wavelengths, high resistance to photobleaching and multicolor fluorescence emission with a single excitation. Thus, QDs are widely utilized in various applications, such as drug delivery, fluorescent biosensing and tissue imaging. Furthermore, QDs have been utilized as promising candidates against amyloidosis formonitoring and inhibiting Aβ aggregation [164].

QDs (NAC-QDs) capped with N-acetyl-L-cysteine were reported to quench both the nucleation and elongation processes resulting from the intermolecular attractive interactions, such as hydrogen bonding between NAC-QDs and amyloid fibrils and the blockage of the active elongation sites on the Aβ fibrils [165]. Furthermore, NAC-QDs have a neuro-protective ability against the cytotoxicity induced by Aβ peptides on human neuroblastoma SH-SY5Y cells [166]. Dihydrolipoic acid(DHLA)-capped CdSe/ZnS QDs also reduced the fibrillation process [167].

### 2.7. Self-Assembled Nanomaterials

#### 2.7.1. Liposomes

Liposomes have been used in drug delivery, because of their non-toxic, high drug-loading capacity and their ease of preparation and modification. In 1999, Maria et al. studied the structure of Aβ_25–35_ and explored its association with different phospholipid membrane vesicles [168]. It was found that three kinds of negatively charged vesicles could accelerate the aggregation of Aβ_25–35_ based on the electrostatic interaction, while vesicles formed by the zwitterionic phospholipid could slow down the aggregation of Aβ_25–35_. Neutral liposomes increase the time of Aβ aggregation in a concentration-dependent manner [169]. The effect of NLs with different sizes on the Aβ aggregation was also investigated by Terakawa and co-workers [170]. Liposomes with smaller sizes (<50 nm) promoted the nucleation and yet those with larger sizes decreased the amount of fibrils and had no influence on the lag time of fibrillation. Shimanouchi et al. reported that Cu^2+^ affected the fibrillar aggregates formed on the surface of oxidized and negatively charged liposomes, such as the oxidatively damaged neuronal cell membranes [171]. Thus, anionic liposomes can result in the formation of spherulitic Aβ aggregates.

Various methods have been proposed by incorporating or modifying the liposomes with different molecules, peptides or antibodies for targeting the Aβ aggregates and plaques [172,173,174]. Nanoliposomes containing anionic phosphatidic acid (PA) or cardiolipin (CL) can bind with all formats of Aβ_1–42_ aggregates with high affinity and thus reduce Aβ-induced toxicity [175,176]. Mourtas et al. found that the planarity of curcumin on the liposome has an important influence on the affinity toward Aβ aggregates, which is dependent upon the conjugation method [177]. Moreover, Taylor et al. demonstrated that curcumin-modified liposome synthesized using click chemistry was the most effective in the inhibition of Aβ aggregation [178]. Canovi et al. decorated NLs with an anti-Aβ monoclonal antibody (Aβ–MAb) to achieve a high affinity toward Aβ monomers and fibrils [179]. Moreover, the multifunctional conjugation of NLs containing PA, CL, curcumin with apolipoprotein E or the anti-transferrin receptor antibody can facilitate the crossing of the BBB and enhance the uptake in the brain capillary cells without the sacrifice of Aβ targeting [180,181,182]. For instance, liposomes bi-functionalized with PA and an ApoE-derived peptide destabilized the preformed Aβ_1–42_ aggregates under the synergic action and could cross the BBB in vitroandin vivo [183,184].

#### 2.7.2. Polymer Nanoparticles

Celia et al. demonstrated that copolymeric NiPAM:BAM nanoparticles increased the nucleation time of Aβ fibrillation, but the elongation step remained largely unaffected, which isdependent uponthe concentrationand hydrophobicity [185]. Through studying the effect of cationic amino-modified PS NPs, they indicated that there is a balance between two different pathways: fibrillation of the free monomer in solution and the nucleation and fibrillation accelerated at the particle surface, which can be determined by the ratio between the peptide and NPs concentration (Figure 14A) [186]. Biopolymeric chitosan-based NPs were also reported to show the ability to inhibit Aβ aggregation and disintegrate the preformed fibrils [187]. 

The positively charged fluorescent conjugated polymer NPs (CPNPs) were prepared to inhibit Aβ_1–40_ peptide fibrillation (Figure 14B) [188]. Moreover, CPNPs with excellent photophysical properties provided fluorescence signals for probing the interaction with Aβ peptides. They found that CPNPs could not only inhibit the aggregation of Aβ but also bind with the terminal of seed fibrils, preventing further fibrillation. A photosensitive polymer nanodot was designed by modifying it with a photosensitizer for efficient suppression of Aβ aggregation [189]. Dou et al. produced fluorogenic “nanogrenades” based on super-molecular assembly between organic dyes and conjugated polymers [190]. The quenched fluorescence of dyes in the nanogrenades was recovered after binding with hydrophobic Aβ fibril plaques. The conjugated polymers in the nanogrenades could generate ROS to destruct the Aβ plaques usingwhite light irradiation.

NPs assembled using conjugated polymers can also be applied to construct stimuli-responsive drug delivery systems. Recently, to ensure targeting and selectivity, Lai et al. designed versatile NPs with a high Aβ-binding affinity, stimuli-responsive drug release and a photothermal degradation ability for the dissolution of Aβ [191]. As shown in Figure 14C, the NPs were composed of an NIR-absorbing conjugated polymer-formed photothermal core and a thermal-responsive polymer-formed shell as NIR-stimuli gatekeeper. Inhibitor curcumin was loaded into the NPs andthe peptide LPFFDwas modified on the surface of NPs for targeting Aβ. Upon NIR laser irradiation, local heat generated by the core could not only trigger the release of encapsulated curcumin to inhibit the aggregation of Aβ but also effectively dissociate the Aβ deposits. Moreover, the Aβ fibrillation and disassembly could be real-time monitored due to the intrinsic polarity-dependent fluorescence of curcumin.

Dopamine can self-assemble into melanin-like poly(dopamine) (PDA) NPs under alkaline conditions with oxygen as the oxidant. PDA NPs with functional groups (i.e., catechol and amine) on the surface can interact with peptides and proteins. Our group was the first to find out that PDA NPs could prevent the formation of Aβ fibrils via the hydrogen bonding and aromatic interactions between Aβ and PDA NPs [192]. Moreover, eumelanin-like particles and pheomelanin-like particles could also perturb the Aβ_1–42_ aggregation and remodel the matured Aβ_1–42_ fibers [192].

Inspired by the self-assembly of biomolecules into complicated functionalized units in cells and nature, researchers put efforts into the self-assembly of different molecules from natural small molecules to peptides, even to proteins. Among those self-assembly blocks, low-molecular-weight peptides have attracted significantattention due to their flexible sequences and biodegradability. Recently, Liu et al. proposed a peptide-based porphyrin supramolecular self-assembly (PKNPs) for target-driving the selective photooxygenation of Aβ [193]. Porphyrin-peptide conjugate (PP-KLVFF) can be self-assembled into PKNPs via hydrophobic interactions and π-π stacking interactions, resulting in the suppression of the intrinsic fluorescence emission, the generation of ROS by free porphyrin and the enhancement of photo-to-thermal conversion ability. The photothermal effect facilitated the crossingof the BBB and then Aβ selectively initiated the disassembly of PKNPs into free porphyrin to produce ROS under light irradiation and thus oxygenated the Aβ.

### 2.8. Others

In the PDT, visible (or UV) light-activated photosensitizers are always confronted with the problem that the penetration depth of UV–visible light in biological tissues is limited. To solve this problem, the upconversion nanoparticles (UCNPs) with the ability to convert NIR light into short-wavelength light are attractive for therapeutic applications. As shown in Figure 15A, Kuk et al. proposed a NIR-light-responsive strategy for the suppression of Aβ aggregation [194]. In this work, rattle-structured organosilica-shell-coated, Yb/Er-co-doped NaYF_4_ NPs were synthesized with an interior cavity to encapsulate numerous rose bengal (RB) molecules with a high loading efficiency and no self-aggregation. Since the absorption of RB partially overlapped with the green emission of UCNPs, visible-light-absorbing RB was activated by UCNPs under 980 nm NIR light, through the highly efficient energy transfer to generate oxidative ^1^O_2_, oxidize peptides and preclude the Aβ fibrillogenesis. However, in dark conditions, a delayed elongation rate but an unaltered amount of total Aβ aggregate was recorded, which wasascribed to the intrinsic inhibition ability of the positively charged UCNPs. Moreover, the biocompatible UCNPs also showed effective suppression of the Aβ-induced cytotoxicity under NIR light.

Polyphenol compounds can inhibit Aβ aggregation and decrease the generation of ROS. However, excess metal ions in Aβ plaque can bind with them, resulting in a decreasein efficacy. Ma et al. proposed a NIR-responsive UCNPs-caged system to sequentially release drugs by regulating them using an NIR laser [195]. As shown in Figure 15B, the metal chelator CQ and polyphenol curcumin were conjugated on the surface of UCNPs using two NIR-sensitive linkers. After being irradiated by a low-power laser, CQ was released to remove free metal ions. Then, curcumin was released to clear superfluous ROS by increasing the intensity of the laser, leading to enhanced treatment efficacy.

Chiral amino acids or peptides conjugated on the NPs can endow NPs with chiral properties, which have aroused interest in the applicability of chiral NPs in different fields. Recently, Zhang et al. found that D-type penicillamine (D-Pen)-modified Fe_x_Cu_y_Se nanoparticles (NPs) showed higher efficiency in the inhibition of Aβ_1–42_-monomer aggregation and enhancement of the dissociation of Aβ_1–42_ fibrils under NIR light irradiation in 10 min (Figure 15C) [196].

Sulfur nanomaterials have the excellent ability to remove Cu^2+^ ions and radicals. Sun et al. synthesized three RVG-peptide-modified sulfur NPs (SNPs) with different morphologies and study their influence on the aggregation of Aβ-Cu^2+^ complexes and corresponding neurotoxicity [197]. They found that the sphere-like SNPs exhibited the most effective inhibition activity owing to the small size, thus reducing the Aβ-Cu^2+^-induced ROS and increasingthe cell viability (Figure 15D).

## 3. Conclusions

In this review, we give a brief overview of recent achievements of nanomaterial-based modulation of Aβ aggregation. Nanomaterials for the treatment of AD play multiple roles. First, most nanomaterials can directly interact with Aβ peptidesto accelerate or slow the aggregation. Second, nanomaterials can act as nano-carriers for loading of various drugs to allow themto cross the BBB and improve the local concentration of drugs. Third, nanomaterials with photosensitive properties can influence the format of Aβthrough PTT or PDT. Although the development of nanomaterials opens a brandnew chapter in the treatment of AD, more efforts are still urgently needed and more novel nanomaterials should be explored and investigated. For example, MXene, which is a novel type of 2Dnanosheet that is mainly composed of early transition metal carbides, has attracted a great deal of attention in energy evolution and nanomedicine [198]. Moreover, metal ions on the MXene surface may interact with Aβ peptides or aggregates. Although there are increasing studies focusing on the interactions between nanomaterials and Aβ, a deep and comprehensive understanding of its nature and application is still necessary.

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
