# Peer review of "Nanomaterials for Modulating the Aggregation of β-Amyloid Peptides"

_molecules, 2021, doi:10.3390/molecules26144301_

Round 1

Reviewer 1 Report

The manuscript by Yaliang et. Entitled "Nanomaterials for modulating the aggregation of β-amyloid peptide", it covers important aspects for the scientific community, but I believe that some aspects need to be shown for the reader to understand: - A summary table would need to be shown with the main advantages and disadvantages of nanomaterial - A graphic summary with the important characteristics of each nanomaterial for the purpose of the manuscript would need to be prepared. These two pieces of information are very important for understanding the article.

Author Response

We thank the reviewer for his/her positive comments. A graphic summary has been added in the revised manuscript, as shown in Scheme 1.

Reviewer 2 Report

In this manuscript, the authors suggested an over-view of recent efforts to utilized nanomaterials with superior properties to propose effective platforms for AD treatment. This manuscript will be of interest to the general readers of molecules because this review described a over-view of recent development of nanomaterials. In addition, this manuscript has provided a lot of information and direction to many researchers studying Alzheimer`s treatment and diagnosis. The manuscript is ready for publication.

Author Response

We thank the reviewer for his/her positive comments. We have checked the manuscript carefully and revised the spelling mistakes.